# Genetics of Ataxias in Indian Population: A Collative Insight from a Common Genetic Screening Tool

*Pooja Sharma, Akhilesh Kumar Sonakar, Nishu Tyagi, Varun Suroliya, Manish Kumar, Rintu Kutum, Vivekananda Asokchandran, Sakshi Ambawat, Uzma Shamim, Avni Anand, Ishtaq Ahmad, Sunil Shakya, Bharathram Uppili, Aradhana Mathur, Shaista Parveen, Shweta Jain, Jyotsna Singh, Malika Seth, Sana Zahra, Aditi Joshi, Divya Goel, Shweta Sahni, Asangla Kamai, Saruchi Wadhwa, Aparna Murali, Sheeba Saifi, Debashish Chowdhury, Sanjay Pandey, Kuljeet Singh Anand, Ranganathan Lakshmi Narasimhan, Sanghamitra Laskar, Suman Kushwaha, Mukesh Kumar, Cheruvallill Velayudhan Shaji, Madakasira Vasantha Padma Srivastava, Achal K. Srivastava, Mohammed Faruq,\* and GOMED-Ataxia study group*

Cerebellar ataxias (CAs) represent a group of autosomal dominant and recessive neurodegenerative disorders affecting cerebellum with or without spinal cord. Overall, CAs have preponderance for tandem nucleotide repeat expansions as an etiological factor (10 TREs explain nearly 30–40% of ataxia cohort globally). The experience of 10 years of common genetic ataxia subtypes for ≈5600 patients' referrals (Pan-India) received at a single center is shared herein. Frequencies (in %, n) of SCA types and FRDA in the sample cohort are observed as follows: SCA12 (8.6%, 490); SCA2 (8.5%, 482); SCA1 (4.8%, 272); SCA3 (2%, 113); SCA7 (0.5%, 28); SCA6 (0.1%, 05); SCA17 (0.1%, 05), and FRDA (2.2%, 127). A significant amount of variability in TRE lengths at each locus is observed, we noted presence of biallelic expansion, co-occurrence of SCA-subtypes, and the presence of premutable normal alleles. The frequency of mutated GAA-FRDA allele in healthy controls is 1/158 (0.63%), thus an expected FRDA prevalence of 1:100 000 persons. The data of this study are relevant not only for clinical decision making but also for guidance in direction of genetic investigations, transancestral comparison of genotypes, and lastly provide insight for policy decision for the consideration of SCAs under rare disease category.

P. Sharma, N. Tyagi, M. Kumar, R. Kutum, V. Asokchandran, S. Ambawat, U. Shamim, A. Anand, B. Uppili, A. Mathur, S. Parveen, S. Jain, J. Singh, M. Seth, S. Zahra, A. Joshi, D. Goel, S. Sahni, A. Kamai, S. Wadhwa, A. Murali, S. Saifi, M. Faruq
Genomics and Molecular Medicine
CSIR-Institute of Genomics and Integrative Biology (CSIR-IGIB)
Mall Road, Delhi 110007, India
E-mail: faruq.mohd@igib.res.in; faruq.mohd@igib.in

P. Sharma, N. Tyagi, M. Kumar, V. Asokchandran, B. Uppili, S. Zahra, A. Kamai, S. Wadhwa, M. Faruq
Academy for Scientific and Innovative Research
Ghaziabad, Uttar Pradesh 201002, India

A. K. Sonakar, V. Suroliya, I. Ahmad, S. Shakya, J. Singh, M. V. P. Srivastava, A. K. Srivastava
Neurology Department
Neuroscience Centre
New Delhi 110029, India

D. Chowdhury, S. Pandey
Department of Neurology
GB Pant Hospital
Delhi 110002, India

K. S. Anand
Department of Neurology
Post Graduate Institute of Medical Education and Research
Dr. Ram Manohar Lohia Hospital
New Delhi 110001, India

R. L. Narasimhan
Institute of Neurology
Madras Medical College
Chennai 600003, India

S. Laskar
Department of Neurology
Safdarjung Hospital
Delhi 110029, India

## 1. Introduction

Ataxia disorders which are inherited in an autosomal dominant fashion are commonly referred as spinocerebellar ataxias (SCAs). The term spinocerebellar although implies impairment of both spinal cord and cerebellum, however the involvement is predominantly of cerebellum and not necessarily the clinical semiology does not always correspond to lesions of the spinal cord. Various other brain regions such as basal ganglia, peripheral nerves, brainstem etc. are also affected. To date, there are numerous genetically distinct varieties of SCAs, SCA1-48 chiefly, and loci exhibiting tandem repeat nucleotide expansion represent the more common types of ataxias (SCA1, SCA2, SCA3, and SCA6 globally). From the experience of various neurology clinics and research efforts, the identification of definitive genetic etiology remains in the range of (20–40%) and a larger proportion of cases remain unknown for their etiological diagnosis.[1,4] The prevalence figures remained unknown at population scale for SCAs; however it is estimated to be 2.7 in 100 000 individuals.[4] The reported incidents of each SCA subtype vary in different geographical regions and ethnic populations; however, SCA3/MJD is considered the most common subtype worldwide. As per reported frequency, SCA2 is most common in Cuba, India and Mexico, while SCA1 is more common in European countries, SCA6 being common in Japan and Germany and various other Caucasian populations making it third common ataxia type across the globe. Similarly, among specific SCA subtypes, prevalence of SCA7 is higher in Scandinavia and Venezuela, SCA12 in India and SCA17 is one of the least prevalent SCA worldwide (in polyglutamine SCAs) with frequency of 0–3% of SCA families.[1,5–7] In addition the SCA subtypes linked to trinucleotide repeat expansion (TRE), the SCAs (SCA5, SCA11, SCA15, etc.) caused by single nucleotide mutations (SNMs) are less frequently reported and this fact could be biased owing to the technical challenge in investigating the SNMs per gene in these rare SCAs in comparison to investigating one mutation per gene linked to TRE loci.[2,3]

In the Indian context, in the last two decades, multiple efforts from different academic hospitals, research settings and commercial settings are in place who routinely investigate the SCA-TRE loci as a primary work-up for deciphering the etiology. Overall, the reported occurrence of SCA subtypes, i.e., SCA1, SCA2, SCA3, and SCA12 is common across the country, whereas, few cases reports have also described SCA6, SCA7, SCA10, SCA17, and DRPLA.[8–12,58]

It is anticipated that from India, the data regarding the occurrence of SCA subtypes is under-represented than actual known data to the health care settings. Thus it is imperative to put the data in the public domain for understanding the global perspectives about SCAs. Among various genetic testing services in India, our center in collaboration with pan-India network of clinicians under GOMED (Genomics and other Omics tools for Enabling Medical Decisions, http://gomed.igib.in/) program caters to the need of health care setting for research based diagnostic services for various genetic ailments including SCAs since 2015. Here, with this study, we share our experience from investigations of SCAs and FRDA linked to TRE mutations in the largest cohort (≈5600) till date from India.

## 2. Results

### 2.1. General Findings

A total of 5594 samples (patients: 4604; family members: 990) suspected of ataxia were recruited in our study in a time span of 10 years (2010–2021) aging from 4 months to 94 years (average = 41.05 ± 18.30 years). Of all samples screened, $n = 197$ (patients: 160; family members: 37) were aged 10 years or below. Out of all individuals, 66.3% ($n = 3712$) were males and 33.6% ($n = 1884$) were females. The frequency of occurrence of the disease in male and female is 18.33% and 9.68%, respectively. This might be due to recruitment of less number of females in the cohort.

Patient recruitment was based on presentation of cerebellar ataxia associated manifestations at examination. Around 26.8% of cases were diagnosed positive for various SCA types and FRDA. The frequency of different SCA types in our sample cohort is SCA12 (8.6%), SCA2 (8.4%), SCA1 (4.8%), SCA3 (2%), SCA7 (0.5%), SCA6 (0.1%), and SCA17 (0.1%) while 73.2% remain genetically uncharacterized for screened loci (**Figure 1**A). Higher frequency of SCA12 and SCA2 than other SCAs indicates higher prevalence of these particular SCA types in the Indian population. For cases referred with autosomal recessive cerebellar ataxia (ARCA) and autosomal dominant cerebellar ataxia (ADCA) phenotypes with age less than 30 years ($n = 822$), the estimated frequency of FRDA in the sample cohort is 15.45%. The overall calculated frequency of FRDA among all studied ataxia types is around 2.2% from our data (Figure 1A). Of the recruitments aging 10 years or below, $n = 20$ samples tested positive for SCAs and FRDA. Of these 20 samples, $n = 9$ had TRE in FRDA gene, $n = 6$ tested positive for SCA2, $n = 2$ had expansion mutations for SCA3 and SCA7 each. One sample was screened positive for SCA1 expansion mutation.

The samples were screened according to the screening strategy as described in the Experimental Section. Genetic screening varied for samples with reference to the clinical context. Additionally, the genetic panel underwent modifications over the years to expand the range of genetic screening offered in accordance with findings relating to SCA prevalence and samples were screened according to the genetic panel available at the time.

### 2.2. Age Distribution of Patients

Mean Age of Examination (in years) of recruited cases has been listed in **Table 1**. On evaluation of SCA positive patients' ages (at examination), it is observed that SCA2 is prevalent in younger population (18–29 years); SCA12 in elder age group (50–59) and other SCA types in middle age group (30–49) (**Figure 2**).

S. Kushwaha
Department of Neurology
Institute of Human Behaviour and Allied Sciences
Delhi 110095, India

M. Kumar
Max Superspeciality Hospital
Delhi 110017, India

C. V. Shaji
T. D. Medical College
Vandanam
Alappuzha, Kerala 688005, India

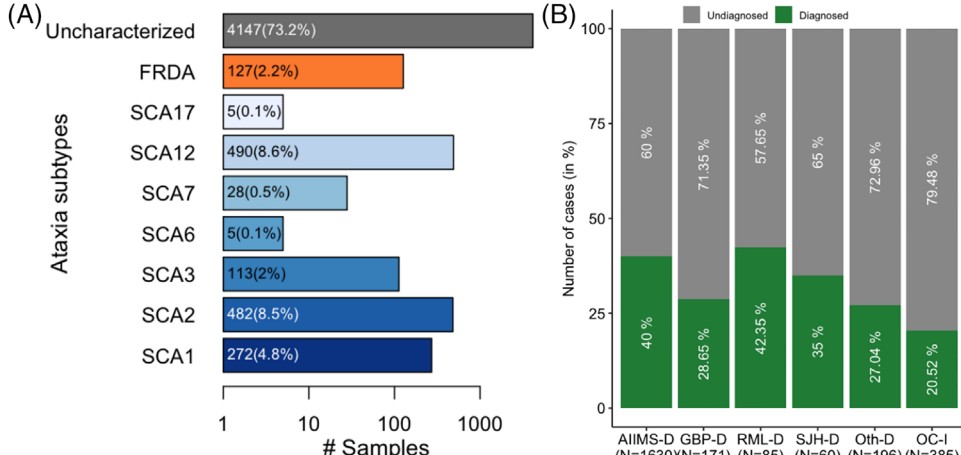

**Figure 1.** Genetic spectrum of cerebellar ataxias in Indian population: A) bar plot representing frequency of ataxia subtypes in our sample cohort ($n$ = 5594); *uncharacterized: cases carrying no mutations in studied ataxia subtypes B) A between center comparisons of genetic diagnosis established for patients groups from major clinical centers (GOMED-Ataxia Study Group) (2015–2020 study period), where X-axis: referral centers, $n$ = number of referrals. Y-axis: number of cases in percentage, diagnosed versus undiagnosed.

## 2.3. Cases of Biallelic expansions and Co-Occurrence of CAG expansions in Two SCA Loci

Out of all the subjects investigated, 19 were detected with biallelic expansions. SCA12 is the most frequent with eight such cases where four were homozygous with CAG-(46/46, 48/48, 55/55, and 56/56) and four were compound heterozygous with CAG-(49/50, 50/51, 45/52, and 48/56) genotype. For SCA7, we diagnosed five cases with biallelic expansions, two were homozygous with CAG-(42/42 and 47/47), and three of them were compound heterozygous with CAG-(40/41, 45/46, and 51/52) repeats. For SCA2, a total of three biallelic expansions were found in which one was homozygous with CAG- 32/32 and others were carrying CAG-(34/35 and 34/40) expansion in compound heterozygous state. For SCA1, two cases were identified. One with CAG-48/48 and other is compound heterozygous with CAG-43/54. For SCA6, we report a single homozygous case with CAG-21/21. No biallelic expansions were diagnosed in SCA3 and SCA17 (Table 1).

Interestingly, we observed three cases with trinucleotide expansion mutation in more than one SCA genes, two of which showed coexistence of expanded alleles in SCA1 (CAG-30/40, 29/42) and SCA2 (CAG-23/45, 23/41), the other one was diagnosed with co-occurrence of SCA2 (CAG-22/34) and SCA12 (CAG-9/44) expansion mutation.

## 2.4. Frequencies of Premutable Normal Alleles (PMNAs)

We calculated the cut-off values of premutable normal allele in our cohort (Table 1) in uncharacterized and normal allele of positive SCA cases. Surprisingly, *ATXN1* leads the data with highest frequency (10.6%) of PMNAs contradicting the higher prevalence of SCA2 and SCA12 in North Indian Population. The frequency is followed by *ATXN7*: 6.09%, *TBP*: 5.59%, *ATXN2*: 5.03%, *CACNA1A*: 4.28%, *PPP2R2B*: 2.68%, and *ATXN3*: 2.42% with respective cutoff values in uncharacterized ataxia cases. The

PMNAs frequency of ATXN7 is highest (7.14%) in normal alleles of positive cases therefore contributing a possible explanation for biallelic expansion in 5 cases. The distribution of PMNAs among age groups revealed the majority in 18–29 years in SCA3, SCA6 and SCA12 as depicted in **Figure 3**. The other age groups that contain major PMNAs are 40–49 years (SCA1, SCA7) and 30–39 years (SCA2 and SCA17) (Figure 3).

We came across cases that were positive for one SCA type but also exhibit borderline CAG repeats for other different SCA types. In SCA1 positive cases, one case exhibited borderline repeats number in SCA2 (CAG-31) and one other patient in SCA6 loci (CAG-17). Similarly, In SCA2 positive patients, 3 cases had borderline repeats in SCA1 (CAG-36 ($n$ = 2), 37 ($n$ = 1)) and one in SCA3 (CAG-43) where in SCA3 positive patients, we came across one case of borderline SCA1 (CAG-37) only. There are 3 SCA12 positive cases with borderline CAG repeats in SCA1 (CAG-36), SCA6 (CAG-17), and SCA17 (CAG-47) loci.

## 2.5. Clinical Findings

As mentioned, patients recruited were referred by different clinical centers across India. Initially, recruited patients were majorly referred from All India Institute of Medical Science, New Delhi, from 2010 to 2014 for research studies. To prevent any unknown biases, in Figure 1B, we included referral data in reference to GOMED-Ataxia Study Group (2015–2021). Major referral center was All India Institute of Medical Sciences, New Delhi (AIIMS-D) ($n$ = 1630) followed by GB Pant, New Delhi (GBP-D) ($n$ = 171); Ram Manohar Lohia, New Delhi (RML-D) ($n$ = 85); and Safdarjung Hospital, New Delhi (SJH-D) ($n$ = 60). Different other referrals from New Delhi were combined as Others, New Delhi (Oth-D) ($n$ = 196). Samples received from different Indian states were categorized as Other Centers of India (OC-I) ($n$ = 385).

Among the referrals, it is observed that AIIMS-D and RML-D are efficient in correct clinical diagnosis of the genetic ataxia types with an estimated 40–42% genetically diagnosed cases

**Table 1.** General demographics of studied SCA subtypes.

| SCA disorder (pathogenic CAG repeat length) | Gene | Genetically uncharacterized cases | | | Positive cases | | | | | |
| --- | --- | --- | --- | --- | --- | --- | --- | --- | --- | --- |
| | | Normal CAG: mean ± SD, median (range) | Commonest CAG repeat number (mode) | Premutable normal range frequency (CAG cut off) | Normal CAG: mean ± SD, median (range) | Premutable normal range frequency (CAG cutoff) | pathogenic CAG: mean ± SD, median (range) | Age of examination in years: mean ± SD (range) | Biallelic expansion cases | Total positives (N), male/female |
| SCA1 (≥39) | ATXN1 | 28.96 ± 2.05, 29 (20–38) | 29 | 10.6% (33) | 28.87 ± 2.21, 29 (21–36) | 5.88% (33) | 50.7 ± 6.18, 50 (39–70) | 36.11 ± 11.59 (6–73) | 2 | 272 (192/80) |
| SCA2 (≥32) | ATXN2 | 22.25 ± 1.04, 22 (7–31) | 22 | 5.03% (24) | 22.3 ± 1.02, 22 (14–31) | 4.34% (24) | 42.32 ± 4.88, 41 (32–60) | 34.95 ± 13.29 (6–78) | 3 | 482 (314/168) |
| SCA3 (≥45) | ATXN3 | 23.22 ± 5, 24 (10–41) | 24 | 2.42% (33) | 23.17 ± 4.62, 24 (14–38) | 0.87% (33) | 72.49 ± 4.89, 73 (45–89) | 37.48 ± 12.18 (2–64) | 0 | 113 (71/42) |
| SCA6 (≥20) | CACNA1A | 11.1 ± 2.02, 11 (4–18) | 13 | 4.28% (14) | 11.5 ± 1.29, 12 (10–13) | 0 | 21.2 ± 0.54, 21.5 (21–22) | 47 ± 6.63 (40–57) | 1 | 5 (4/1) |
| SCA7 (≥37) | ATXN7 | 10.46 ± 1.56, 10 (8–36) | 10 | 6.09% (14) | 10.73 ± 1.35, 11 (9–14) | 7.14% (14) | 47.81 ± 9.30, 45 (38–76) | 35 ± 14.66 (3–65) | 5 | 28 (20/8) |
| SCA12 (≥43) | PPP2R2B | 12.19 ± 3.12, 12 (6–42) | 13 | 2.68% (20) | 13.13 ± 4.6, 13 (7–42) | 3.89% (20) | 55.01 ± 5.43, 55 (43–73) | 54.5 ± 11.15 (13–85) | 8 | 490 (313/177) |
| SCA17 (≥49) | TBP | 37.26 ± 1.96, 37 (28–48) | 38 | 5.59% (41) | 37.8 ± 0.83, 38 (37–39) | 0 | 70 ± 32.90, 56 (53–129) | 32.8 ± 13 (22–54) | 0 | 5 (4/1) |

Listed SCAs are repeat expansion disorders where expansion of a trinucleotide repeat (CAG) above a particular threshold is the major cause of the disease. Molecular and genetically analyzed repeat lengths for positive as well as unknown cases are shown in Figure 2. General characteristics of SCAs in our sample cohort are listed in Table 1.

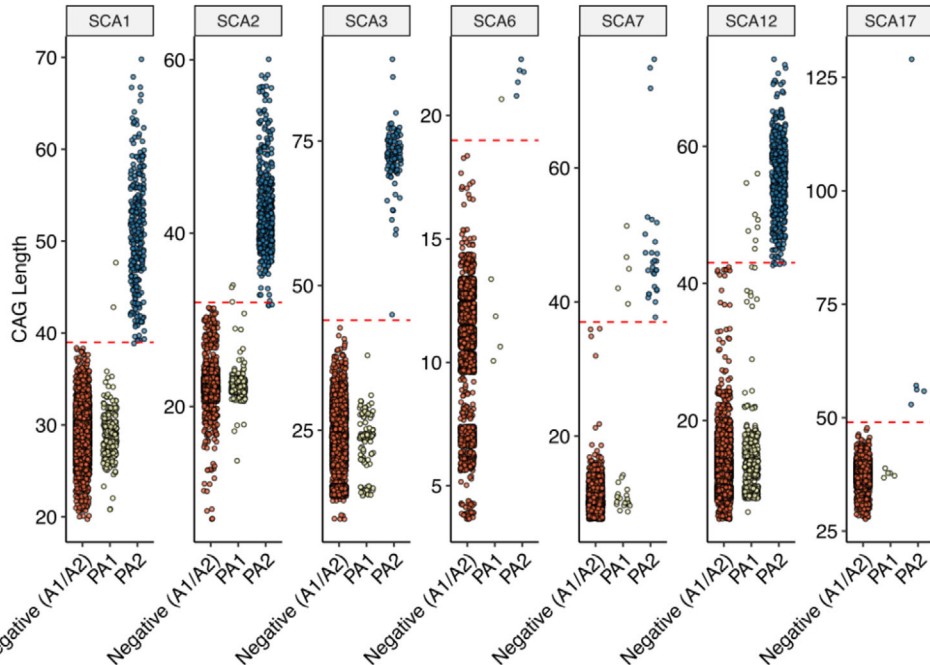

**Figure 2.** Spinocerebellar ataxia (SCA) subtype specific plot for CAG length distribution in Indian CA cohort. In each plot, *X*-axis represent, negative (A1/A2): combined distribution of CAG-allele-1 and CAG-allele-2 in subjects who were negative for CA-panel; PA1 and PA2: SCA-CAG allele-1 and, SCA-CAG allele-2 in subjects who were positive for that specific SCA-subtype. *Y*-axis depicts the CAG repeats length observed in subjects negative or positive for SCA subtype. The red dotted line depicts the threshold of the pathogenic-CAG length for the respective SCA type. *n* = number of samples genetically screened for each SCA type [SCA1 (*n* = 5156); SCA2 (*n* = 5201); SCA3 (*n* = 5182); SCA6 (*n* = 2406); SCA7 (*n* = 2402); SCA12 (*n* = 5225); SCA17 (*n* = 2844)].

(Figure 1B). The frequency of SCAs among genetically diagnosed cases in accordance with the referral centers varied. SCA12 is the most prevalent in genetically diagnosed cases of AIIMS-D (14%), RML-D (15.3%), and Oth-D (12.2%). In GBP-D and OC-I, SCA1 had high prevalence with 8.2% and 7.5% of the solved cases respectively whereas cases referred from SJH-D showed high prevalence in SCA2 (15%) (**Table 2**).

Samples were referred to our center with several clinical diagnostics as mentioned in Table S1 (Supporting Information). To understand the efficiency of clinical diagnosis of these referred cases, we calculated the clinical sensitivity for each SCA type. Formula used to calculate sensitivity is as follows: TP/(TP + FN), where TP = true positive (cases diagnosed as a particular SCA type and confirmed genetically) and FN = false negative (clinical diagnosis discordant with genetic testing results). Sensitivity for clinical diagnosis of various SCA types is calculated as follows: SCA1: 29.4%; SCA2: 49.5%; SCA3:28.16%; SCA6: 20%; SCA7: 30.7%; SCA12:74.85%; SCA17: NIL; and FRDA:82.63%.

### 2.6. FRDA Occurrence and Carrier Frequency Estimation in India

We screened *n* = 822 samples (680 patients and 142 family members) from our cohort with mean age 23.07 ± 12.04 years referred with ADCA/ARCA phenotype. Of the total samples screened, 507 were males and 315 were females. The frequency of FRDA in our sample cohort is 15.45% for ARCA and ADCA referrals below age 30 years. The overall frequency of FRDA in our cohort is 2.2% (Figure 1A). The age of examination of the positive patients

ranges from 8 to 43 years (mean age 20.49 ± 7.07 years). Out of all FRDA positive patients, 63.77% were males. Majority of the diagnosed cases were referred from AIIMS-D (39.2% solved) followed by OC-I (16.5%) (Table 2).

FRDA being a progressive ataxia with autosomal recessive inheritance, we studied the carrier frequency estimation in *n* = 790 healthy individual's samples across India. The included individuals did not have a history of any other related disorders. A total of *n* = 384 samples showed homozygous allele status in analysis of flanking PCR. After performing Triplet Primed PCR, we were able to screen *n* = 313 samples successfully, out of which *n* = 05 samples were found to be of positive carrier state with a frequency estimation of 0.63% (1/158). Among the five FRDA carriers, three were males and two were females. According to the Hardy–Weinberg principle, the expected prevalence of the FRDA in the Indian population is $10e^{-6}$ that is 1/100 000. Large normal alleles were observed in 149 individuals with a GAA range 16–28 repeats. Allele frequency for the highest repeat (i.e., $(GAA)_{20}$) in the shorter allele is 0.0922 and in the larger allele (i.e., $(GAA)_{28}$) is 0.0923.

## 3. Discussion

Spinocerebellar Ataxia is an autosomal dominant inherited disorder, genetically described in various types (SCA1–48) with an estimated prevalence of ≈2.7/100 000.[4] In India, prevalence of particular SCA types varies with geographical and ethnic diversity.[13-17]

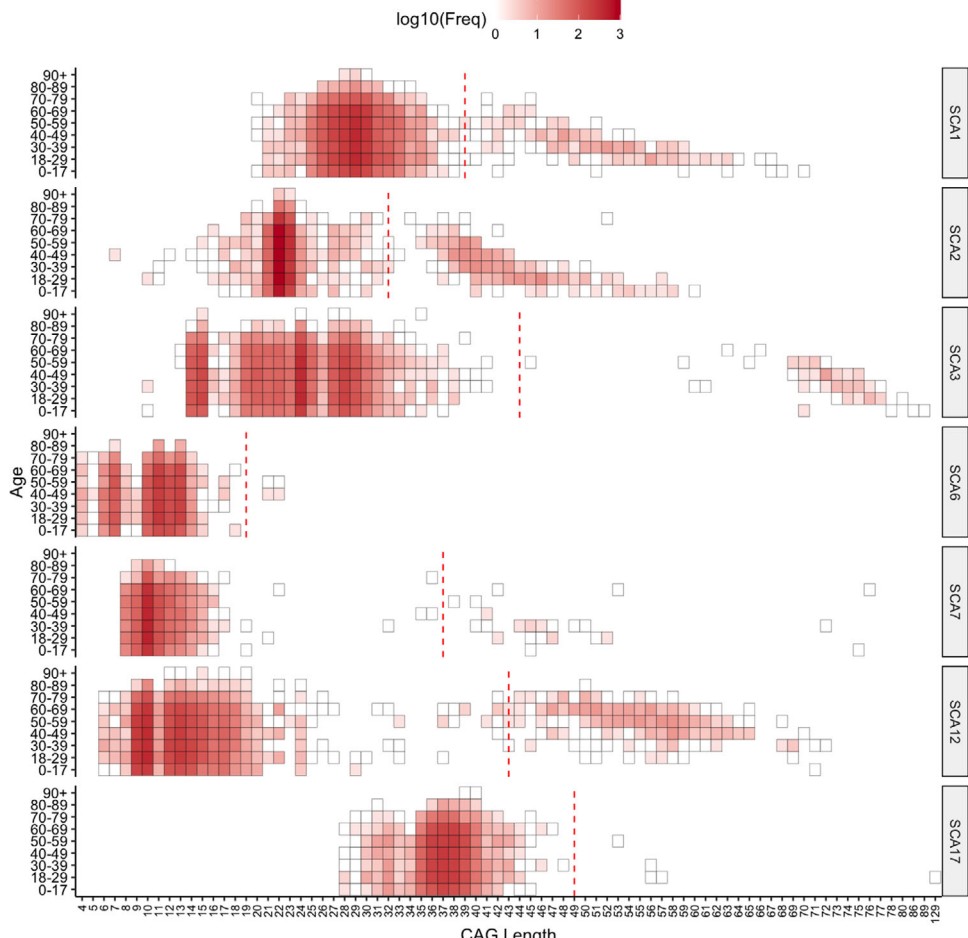

**Figure 3.** Illustrations showing heatmap of age group wise frequency distribution of CAG length (in log10 scale) for each SCA loci. X-axis: CAG- length; Y-axis: age group categories and each box in the heatmap represent the frequency of a specific CAG length in the particular age group. Red dotted line in the plot marks the pathogenic-CAG repeat threshold for each SCA type.

**Table 2.** Percentage of SCA types among genetically diagnosed cases in accordance with the referral centers in India ($^*n =$ total number of positive diagnosed cases).

| | SCA1 | SCA2 | SCA 3 | SCA6 | SCA7 | SCA12 | SCA 17 | FRDA |
|---|---|---|---|---|---|---|---|---|
| REFERRALS | | | | | | | | |
| AIIMS-D ($n = 652$) | 6.1 | 12.9 | 2.2 | 0.1 | 1.0 | 14.0 | 0.2 | 7.5 |
| GBP-D ($n = 49$) | 8.2 | 8.2 | 1.8 | 0.6 | 0.6 | 7.6 | 0.0 | 5.3 |
| RML- D ($n = 36$) | 8.2 | 8.2 | 4.7 | 0.0 | 2.4 | 15.3 | 1.2 | 2.4 |
| SJH-DI ($n = 21$) | 15.0 | 15.0 | 0.0 | 0.0 | 0.0 | 0.0 | 0.0 | 6.7 |
| Oth-D ($n = 53$) | 3.6 | 6.1 | 0.5 | 0.5 | 1.5 | 12.2 | 0.0 | 3.6 |
| OC-I ($n = 79$) | 7.5 | 4.9 | 2.6 | 0.3 | 1.3 | 1.3 | 0.0 | 7.5 |

Currently the first approach to diagnose ataxia is to test TREs, with an estimated 70% of cases remaining uncharacterized. SCAs like SCA5, SCA11, SCA15, etc., are caused by single nucleotide mutations (SNMs) which are less frequent. Efforts are being made by various groups where uncharacterized cases are explored using next generation sequencing to decipher the un-

known genetic cause.[1] Intense NGS-Clinical Exome sequencing has helped improve characterization of ataxia cases, with a yield of ≈50–60%. Further, once an extensive study has been performed, targeted next generation sequencing approach probably helps to evaluate the unsolved cases with cost-effective and limited resources.[33–36]

In our study, we aimed to explore the relative prevalence of spinocerebellar ataxias in the population and understand the efficiency of diagnostics techniques. Indeed, similar studies have been performed earlier in the Indian cohort (**Table 3**) albeit on a smaller scale. Here we share our experience from investigations of SCAs linked to TRE mutations in the largest cohort ($n = 5594$) till date. The observed high prevalence of SCA2 and SCA12 correlated with the peculiar prevalence of mentioned SCAs in north India.[32,68] Remaining 73.2% uncharacterized cases can be explored further to detect SNMs diagnosis using next generation sequencing techniques.

Patients and family members of age less than 10 years were referred either in reference to juvenile onset or were screened for anticipations of CAG repeats in the next generation. Variable onset in spinocerebellar ataxia from infantile to late life has been

**Table 3.** Frequency distribution of SCAs studied by various groups in India (*n* = total number of cases studied; * Studies in which patients were screened only for a particular SCA subtype.).

| Reported publications | Region | n | SCA1 | SCA2 | SCA3 | SCA6 | SCA7 | SCA12 | SCA17 | DRPLA | FRDA | UC-SCA |
|---|---|---|---|---|---|---|---|---|---|---|---|---|
| Current study | India | 5594 | 272 | 482 | 113 | 5 | 28 | 490 | 5 | – | 127 | 4147 |
| Bhanushali et al., 2020[71] | | 700 | 30 | 101 | 32 | 2 | – | 229 | – | – | – | |
| Venkatesh et al., 2018[72] | Bangalore (South) | 864 | 100 | 98 | 40 | – | – | 8 | – | – | 20 | 598 |
| Pulai et al., 2014[13] | Kolkata (East) | 83 | 13 | 18 | 7 | 6 | – | 1 | – | – | – | 38 |
| Netravathi et al., 2009[73] | Bangalore (South) | 24 | 6 | 5 | 3 | – | – | – | – | – | – | |
| Faruq et al., 2009[77] | Delhi (North) | 400 | 27 | 60 | 14 | – | 3 | 45 | – | – | 17 | 234 |
| Krishna et al., 2007[17] | Bangalore (South) | 284 | 39 | 32 | 18 | – | – | – | – | – | – | 202 |
| Alluri et al., 2007[74] | Hyderabad (South) | 124 | 6 | 1 | 1 | – | 1 | – | – | 1 | – | 114 |
| Rengaraj et al., 2005[16] | Chennai (South) | 17 | 17 | – | – | – | – | – | – | – | – | |
| Sinha et al., 2004[9] | Madhya Pradesh, Bihar, West Bengal | 44 | 7 | 26 | – | – | – | – | – | – | – | |
| Wadia et al., 1998[87]; | Mumbai (South-Western) | 6/31 | – | 6 | – | – | – | – | · | – | – | |
| Basu et al., 2000[46]; Chattopadhyay et al., 2003[44]; Sinha et al., 2004[9]; Chakravarty and Mukherjee, 2002[14] | Kolkata (East) | 99 | 12 | 30 | 12 | 1 | – | – | – | – | 6 | 47 |
| Srivastava et al., 2001[75] | Delhi (North) | 77 (families) | 12 | 19 | 2 | – | 2 | 5 | – | – | – | |
| Hire et al., 2011*[76] | Southern India | 481 | – | – | – | – | – | 2 | – | – | – | |
| Sharma et al., 2012[78] | Gwalior, MP (central part of India) | 24 | 5 | – | – | – | – | – | – | – | – | |
| Wali et al., 2013*[79] | Belgaum region of Northern Karnataka, Southern India | 9 | – | – | – | – | 6 | – | – | – | – | |
| Faruq et al., 2014[54] | | 2 | – | 2 | – | – | – | 2 | – | – | – | |
| Singh et al., 2014*[80] | Lucknow, UP | 1 | – | 1 | – | – | – | – | – | – | – | |
| Kumaran et al., 2014*[81] | Adukkamparai village in Vellore, India | 100 | 37 | – | – | – | – | – | – | – | – | |
| Faruq et al., 2015[50] | Haryana (a state in Northern India) | 35 | – | – | – | – | 22 | – | – | – | – | |
| Lone et al., 2016*[82] | Hyderabad, Telangana (Southern India) | 188 | – | – | – | – | – | 1 | – | – | – | |
| Singh et al., 2017*[83] | Lucknow, Uttar Pradesh (Northern India) | 1 | – | 1 | – | – | – | – | – | – | – | |
| Kasinathan et al., 2017*[84] | PGMIR, Chandigarh (North India) | 1 | – | 1 | – | – | – | – | – | – | – | |
| Anjanappa et al., 2019*[85] | Bangalore | 38 | – | – | 18 | – | – | – | – | – | – | |
| Goel et al., 2019[86] | | 461 | – | – | – | – | – | – | – | – | – | |

reported with varying disease progression.[66] Early onset of the spinocerebellar ataxia has also been known to be associated with large CAG expansion mutations in respective genes.[67]

On evaluation of positive patient's age at examination in SCAs, it was observed that SCA2 is prevalent in younger population (18–29 years); SCA12 in the elder age group (50–59) and other SCA types in middle age group (30–49) (Figure 3). Earlier studies have reported occurrence of SCA2 in the third to fourth decade which is significantly late compared to our cohort results. These explorations become fruitful in improving community knowledge, especially among at-risk populations, where asymptomatic carriers can be cautioned about developing the symptoms at observed age.

We identified 19 patients carrying biallelic expansions in various SCAs (SCA1, SCA2, SCA6, SCA7, and SCA12) (Table 1). It has been reported that biallelic expansion of CAG in SCA1 and SCA12 does not cause any clinical or phenotypic severity in comparison to heterozygous expansion with one normal CAG allele.[21,22] Studies reveal in Huntington repeat expansion disorders that homozygous expanded alleles do not cause early age of onset but rather may result in faster disease progression.[23] It has been reported that homozygous borderline intermediate alleles are associated with incidence of the typical disease phenotype in SCA2, SCA6, and SCA17.[19,24,27,28] Expansion in both alleles in SCA3, SCA6, and DRPLA has been proven to be associated with phenotypic severity and earlier age of

onset.[18,25,26,20] No bi-allelic CAG expansion in SCA7 is reported so far.

Three cases were observed with trinucleotide expansion mutation in more than one SCA gene. Two of which showed co-existence of expanded alleles in SCA1 and SCA2 the other one was diagnosed with co-occurrence of SCA2 and SCA12 expansion mutation. Literature explorations corroborated co-occurrence of SCA2 with SCA12;[62] SCA2 with SCA10;[63] SCA3 with SCA2 and SCA17;[64] SCA8 with SCA1, SCA2 and SCA6.[65] To the best of our knowledge, the coexistence of expansion in SCA1 and SCA2 has not been reported. We have communicated a detailed case report of the two patients with expansion in SCA1 and SCA2 elsewhere (unpublished).

### 3.1. Premutable Normal Alleles (PMNAs)

In the present study, we have tried to map the prevalence of studied SCAs and the frequencies of PMNAs in corresponding SCA loci. Similar study has been done by Takano et al. 1998 in which close association between frequencies of PMNAs and prevalent SCAs in Japanese and Caucasian population is exhibited.[30] For SCA1, CAG repeats without CAT interruption in PMNA range might turn abnormal in offspring, while CAG stretch having CAT interruptions are studied to have reduced penetrance even in pathogenic range.[22,29,37] PMNAs listed in our cohort for SCA1, may have CAT interruptions in CAG repeats, as we only performed fragment size analysis, we could not rule out this feature of repeat locus. High frequency of PMNAs in *ATXN1* is a cautionary sign in descendants' generation, as SCA1 is the third prevalent repeat expansion disorder in our cohort. Similar studies in SCA2 revealed stability in repeat expansion due to presence of CAA interruptions in PMNAs and deviation from typical ataxic phenotype to other clinical manifestations like parkinsonism, etc.[38,48,60] Studies have also revealed the risk for amyotrophic lateral sclerosis (ALS) associated with large normal CAG repeats in *ATXN2*. The risk increases with increase in repeat number.[39,40] Shamim et al. 2020 reported 1.5% ALS cases harboring intermediate *ATXN2* CAG repeats in Indian cohort.[43] An extensive study on SCA2 large normal /PMNA has been conducted in Cuba in 2001, to estimate the potential risk of abnormal CAG expansion in the descendant population.[41] In our study, we state PMNAs frequency in uncharacterized cases and in normal allele *ATXN2* positive cases, correlates the high prevalence of SCA2 in India (Table 1). Gan et al. 2015 studied CAG repeats for SCA3 in the Chinese population and reported a frequency of 0.28 for large normal alleles along with a comparison of PMNA frequency in different populations with their study. The reported PMNA frequency for *ATXN3* in the Indian population is 0.02 for >31 CAG repeats,[42] while in our study we report a significantly higher frequency (SCA3 ≥ 33 CAG repeats) in our sample cohort. Studies from Eastern India reported intermediate CAG frequencies of SCA3 to be 0.015 in the pooled population. Chattopadhyay et al. 2003 and Mittal et al.2005 showed association of same haplotype with large ANs and expanded alleles, stating the possibility of large ANs turning into expanded alleles.[44,45] The calculated frequency of PMNAs in our sample cohort for *CACNA1A* is 4.28% (CAG ≥ 14). No PMNAs in normal allele of heterozygous positive cases detected for SCA6. Basu

et al. 2000 studied PMNAs frequency for SCA1, SCA2, and SCA6 in nine ethnic population of eastern India and estimated a frequency of 0.211 (>31 CAG) for SCA1, 0.038 (>22 CAG) for SCA2, and 0.032 (>13 CAG) for SCA6.[46] SCA6 loci has been studied to have a significant frequency of PMNA (>13 CAG) in Brazilian cohort and SCA2 loci (>22 CAG) in a Portuguese cohort.[48] *ATXN7* harboring highest and second highest PMNA frequency in normal allele of positive and uncharacterized cases respectively highlights the increasing prevalence of SCA7. Mittal U et al 2005 described a case from India where a postzygotic de novo TRE at SCA7 locus from an intermediate (CAG-31) allele carrier father is explored.[49] Also, Faruq et al. 2015 analyzed 21 diverse Indian populations for ATNX7-CAG repeats and concluded premutable repeats (28–34) in one of the studied populations.[50] SCA12 PMNA frequency in Portuguese group carrying >15CAG repeats were studied to be 0.03 and compared to Brazilian group of individual having a frequency of 0.05.[48] Considering the data from our cohort, the PMNA frequency does not correlate with the high prevalence of SCA12 in the country. This is in accordance with the assumption that SCA12 does not exhibit much intermediate CAG repeats allele range and instead harbor either normal alleles or pathogenic expansions. A new clinical spectrum was setup by Srivastava et al. 2017; when CAG-43 was described to be differentiate into full penetrance from intermediate alleles.[22] PMNAs in SCA17 showed a frequency of 5.59% (≥41 CAG) in uncharacterized patient's cohort of our study. Related research showed complications with the intermediate allele range in *TBP*. The repeat range 41–48 CAG has been studied with variable penetrance. Individuals harboring 41–42 CAG had been characterized as SCA17 positive whereas some studies reported 43–49 CAG as asymptomatic.[47] Similar to SCA6, we did not discover any PMNA in normal allele of SCA17 patients from our sample cohort.

Overall with the calculated frequency of PMNAs in SCA loci, we emphasize upon the probability of attaining future de novo repeat expansions in descendent populations. Some of the cases may result in reduced penetrance for various SCA loci reported. Such investigations could bring better discernment to clinical as well as genotypic aspects of related disorders. We did our study on patient cohort only, analogous studies on healthy control groups to analyze the risk of developing polyglutamine related diseases in the successor population can help enhance existing knowledge.

Notably, borderline CAG repeats in different SCA loci along with the pathogenic expansions were noticed. The association of borderline repeats with other pathological expansions may have an impact on the phenotype severity, disease progression, age of onset or heterogeneity in symptoms.[51,52,53] Such cases may turn into expansion in different two SCA loci in the next generation. Faruq et al. 2014, explored a case with expansion in SCA2 and SCA12 simultaneously.[54] We recommend detailed familial study for upcoming generations in similar instances.

### 3.2. Clinical Diagnostics Sensitivity of SCA Types

The studied subjects were from all over India and through different clinical centers. Ratio of solved cases was compared in between the referral centers to explore the efficiency of clini-

cal diagnostics. We tried to map the prevalence of SCA types in accordance to the referring centers. SCA 12 is the most prevalent in solved cases of AIIMS-D (14%), RML-D (15.3%), and OC-I (12.2%). In referrals from GBP-D and OC-I, SCA1 had high prevalence with 8.2% and 7.5% solved cases respectively whereas cases referred from SJH-D showed High prevalence in SCA2 (15%). Significant overlapping of clinical phenotypes among different SCAs make it challenging to make stringent diagnosis.[31] On the basis of the clinical diagnostics referred for genetic screening of patient's samples (Table S1, Supporting Information) in our cohort, it is observed that phenotypic characteristics may overlap among different SCA types. 21 cases of SCA1 positives were clinically diagnosed as SCA2. Similarly $n = 15$ and $n = 12$ SCA2 positive patients were clinically diagnosed as SCA12 and SCA1, respectively. Among SCA3 positive cases, ten patients were clinically diagnosed as SCA1 and six patients as SCA2 (Table S1, Supporting Information). These observations led us to conclude that the clinical phenotypes overlap among SCA1, SCA2, and SCA3. Also, overlapping symptoms exist between SCA2 and SCA12 cases.

Efficiency of clinical diagnosis plays a major role in detection of the disease. We calculated the sensitivity for each SCA type to calculate the efficiency of clinical diagnosis of these referred cases. Thus, detailed clinical data can help to diagnose the disease efficiently and also helps to prioritize the gene panel testing for various SCA types.

### 3.3. FRDA: Carrier Frequency and Global Comparisons

FRDA is a progressive early onset autosomal recessive inherited ataxia occurring majorly due to expansion of GAA mutations in FXN gene. High prevalence of FRDA is reported in Europe, the Middle East, South Asia, and North Africa.[55] FRDA is prevalent worldwide with sub-Saharan Africa and Southeast Asia as exceptions. The estimated prevalence of FRDA in Central Europe ranges from 1/20 000 to 1/50 000.[56] With high prevalent status of FRDA in Europe, carrier frequency estimation resulted in a frequency of 1/90 in Caucasian populations. Other studies illustrated carrier frequency of 1/196 in Norway, 1/500 in southern Finland, and 1/100 Northern Finland.[59] The prevalence of the disease in India is yet to be explored. FRDA occurrence of 4% in north Indian and 2.4% in south Indian clinically diagnosed population has been mentioned by Singh et al. in 2010.[57] Of clinically diagnosed sporadic and recessive ataxia, they report a FRDA prevalence of 7% in north Indians and 4.8% in south Indians.[57]

To estimate the prevalence of FRDA in the country, we conducted research on the evaluation of carrier frequency in healthy individual's samples. According to the Hardy–Weinberg principle, the expected prevalence of the FRDA with respect to the calculated carrier frequency (1/158) in the Indian population is $10e^{-6}$ (1/100 000). The insights of large normal alleles in the study points out the instability of the GAA repeats which makes it more prone to expansion in successive generations. Although there is the limitation of small sample size, the studied carrier frequency could give an estimate of the prevalence of the disease in India thus lighting up new aspects related to disease.

## 4. Conclusion

In the end, we conclude that our study is relevant from both clinical as well as genetic aspects by providing phenotypic and genotypic analysis of such a large patient cohort in depth in the Indian population. The data of this study can be used i) to serve as proxy data for frequency of SCAs (in the absence of exact prevalence figure), ii) for the establishment of investigational work-up/flow chart for ataxias in India, iii) for drawing comparative studies with global population data and lastly, and iv) to revise the policy decision for consideration of SCAs under rare disease category.

## 5. Experimental Section

*Participants' recruitment and Genomic DNA Extraction*: This study comprised a total of 5594 participants (patients = 4604; family members: 990) referred for genetic investigations (from 2010 to 2021) of ataxia phenotype. The subjects were recruited from multiple referral sites all across India, mainly All India Institute of Medical Science, New Delhi (for throughout the mentioned period) and other major tertiary referral center from India for last 5 years (2015–2021) under GOMED (Genomics and Other Omics Tools for Enabling Medical Decision) program (http://gomed.igib.in). All the subjects recruited were part of the GOMED-Ataxia study group which aims to decipher clinical-genetic correlation in CAs. For the initial phase, the aim was at the diagnostic part of the participants through a common diagnostic SCA panel (i.e., screening of SCA1, SCA2, SCA3, SCA6, SCA7, SCA12, SCA17, and FRDA). All the patients were evaluated by expert neurologists at respective centers and have received clinical diagnosis of one or the other form of genetic-ataxia (insidious onset and progressive behavior) subtypes [(ADCA phenotype as per Harding's classifications) or MSA-C (multiple system atrophy cerebellar) or ARCA or FRDA] through history given by the patient and/or their relatives followed by neurological examination. Subsequent to the clinical diagnosis, patients were referred for genetic investigations to the CSIR-Institute of Genomics and Integrative Biology. A 3–5 mL of peripheral venous blood sample was sent to CSIR-IGIB. All subjects had given their consent for the participation in this research study. Ethical clearance was obtained from the Institutional Human Ethics Committee of CSIR-IGIB (IHEC). The genomic DNA was extracted using salting out method[61] from peripheral venous blood.

*TNR Length Estimation*: The extracted DNA was subjected for trinucleotide repeats (TNRs) length estimation at respective loci through PCR amplification using fluorescently labeled primers.[70] The PCR product size (fragment length of each of the TNR loci) were analyzed by capillary electrophoresis on ABI 3730XL DNA Analyzer (Thermo Fishers Scientific) and size estimation through Genescan software v 4.0. For FRDA screening GAA expansion detection was performed through triplet repeat primed PCR as described earlier.[69]

*Statistical Analysis*: Statistical analysis was performed on the CAG repeat length information. General characteristics are mentioned as mean ± SD, median, mode and range of the data in Table 1. Comparative analysis was done based on frequencies and percentages. In genetically uncharacterized cases, mutable normal allele cutoffs were designated by taking observations under the two degrees of standard deviation (5%) from the mean of repeat lengths. Mentioned statistical analysis has been conducted with built in formulas using MS-Excel version 2013. Hardy–Weinberg principle was applied for carrier frequency estimation of FRDA in the Indian population.

*Screening Strategy*: A total of $n = 5594$ samples of suspected cerebellar ataxia were genetically screened for TRE mutations. Screening strategy is based on the genetic test referred to after neurological consultation. Initially for a time period of 5 years (2010–2014), patients ($n = 2432$) were screened under a primary screening strategy which involves genotyping of SCA1, SCA2, SCA3, and SCA12 loci. Later to expand the diagnostic capabilities, screening of SCA6, SCA7 and SCA17 loci was introduced along

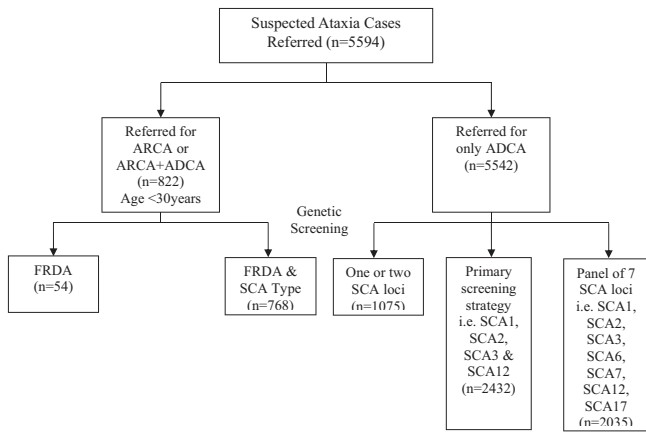

**Figure 4.** Flowchart depicting the screening strategy of referred subjects; *n* = number of samples.

with the primary screening panel (*n* = 2035). For FRDA genotype screening, patients referred with autosomal recessive cerebellar ataxia (ARCA) (*n* = 54) and autosomal dominant cerebellar ataxia (ADCA) phenotypes aged less than 30 years (*n* = 768) were considered. Cases referred to test any specific SCA locus for family based screening (*n* = 1075) were screened for a single SCA locus (as illustrated in the **Figure 4**) to elucidate disorder anticipation.

*Carrier Frequency of Friedreich's Ataxia (FRDA) in Indian Population*: This study included *n* = 790 healthy individuals with mean age 31 ± 17.98 years (1 month to 79 years) and no history of related disorders. GAA repeat sizes genotyping was carried out by flanking PCR with fluorescent labeled primers followed by capillary electrophoresis. Triplet repeat primed PCR was performed to ascertain the carrier status of the samples that showed homozygous allele status after performing flanking PCR. The Hardy–Weinberg principle was applied to find an expected prevalence of FRDA in the Indian population.

## Supporting Information

Supporting Information is available from the Wiley Online Library or from the author.

## Acknowledgements

A full list of the authors/coinvestigators from the GOMED-Ataxia study group is provided in Section S2 in the Supporting information. The authors sincerely thank Suman Mudila, Usha Rawat, and Subhash Gurjar (CSIR-Institute of Genomics and Integrative Biology) for managing the blood and DNA samples. All the patients and their family members who participated and cooperated during clinical and genetic investigations are duly acknowledged by all the authors. This study was part of the genetic characterization aspect of SCA subjects and supported by continuing multiple research projects SIP0006, BSC0123, MLP1601, and MLP1802 funded by CSIR.

## Conflict of Interest

The authors declare no conflict of interest.

## Data Availability Statement

The data that support the findings of this study are available on request from the corresponding author. The data are not publicly available due to privacy or ethical restrictions.

## Peer Review

The peer review history for this article is available in the Supporting Information for this article.

## Keywords

cerebellar ataxias, FRDA, premutable alleles, prevalence, SCA in India

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

www.advancedsciencenews.com

**ADVANCED
GENETICS**

www.advgenet.com

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
