## [**Supplementary Information**: Record of Transparent Peer Review · Advanced Genetics]

Record of Transparent Peer Review

Genetics of ataxias in Indian population: A collative insight from a common genetic screening tool

Pooja Sharma, Akhilesh Kumar Sonakar, Nishu Tyagi, Varun Suroliya, Manish Kumar, Rintu Kutum, Vivekananda, Sakshi Ambawat, Uzma Shamim, Avni Anand, Ishtaq Ahmad, Sunil Shakya, Bharathram Uppili, Aradhana Mathur, Shaista Parveen, Shweta Jain, Jyotsna Singh, Malika Seth, Sana Zahra, Aditi Joshi, Divya Goel, Shweta Sahni, Asangla Kamai, Saruchi Wadhwa, Aparna Murali, Sheeba Saifi, Debashish Chowdhury, Sanjay Pandey, KS Anand, R Lakshmi Narasimhan, Sanghamitra Laskar, Suman Kushwaha, Mukesh Kumar, CV Shaji, MV Padma Srivastava, Achal K Srivastava, Mohammed Faruq* and GOMED-Ataxia study group

*Corresponding

Review timeline:	Date Submitted:	22-Dec-2021
	Editorial Decision:	31-Jan-2022
	Revision Received:	11-Feb-2022
	Accepted:	11-Feb-2022

Editor: Myles Axton

1st Peer Review

4-Jan to 31-Jan-2022

Reviewer #1

[No specific comments for revision]

Reviewer #2

Sharma et al. present the results of a large, multi-center study of SCAs across India. The authors identified eight different subtypes in the population studied, SCAs 12, 2, 1 and 3 being the most common. The results are important in the context of India and also on the frequency of SCAs among different populations worldwide. Findings on biallelic mutations and regional differences are also remarkable. A few issues need to be addressed prior to publication though.

2.1 English language needs to be reviewed.

2.2 Please include the meaning of the acronym TRE after it is first mentioned.

The Introduction is informative and addresses the main data needed to understand the relevance of the study.

2.3 Methodology is adequate, except for a couple of points that require clarification: as the samples were collected and analyzed over a period of 11 years, was there any change in genetic testing as time went by and technology and knowledge in the field evolved? Was the genetic panel changed at all over the years?

2.4 Second: please provide a brief summary of statistical methods as they are not explicitly presented.

2.5 Also, please provide the criteria to define "short positive alleles".

2.6 The finding of several cases with biallelic mutations in SCAs is remarkable and findings should be better explored, specially including genotype/phenotype correlation, if available.

2.7 The large difference in frequencies of the most common forms of SCAs is also inevitable, considering the large territorial size of India and the presence of a possible distinct ethnic background for each of these different areas. This topic should be better explored as well, correlating with the ancestral origin in different areas, if this makes sense in the context.

2.8 Finally, FRDA is not considered a form of SCA, therefore the inclusion of data on this ARCA should warrant a change in the manuscript's title.

Reviewer #3

I would like to start by congratulating the authors on such an interesting work. This is a collaborative work, with an impressive number of patients, and very important conclusion in terms of frequency and characterization of SCAs in India. I agree with the authors with this paper being very important for Indian neurologists, in guiding their practice and genetic testing.

I would just have a few comments/doubts

3.0 The paper would benefit from an overall grammar review. There are some very long sentences, difficult to read. A clear and objective language should be preferred.

Results

3.1 (In section 2.1) I couldn't help noticing a patient with 4 months included in this study. This is very unusual, and a comment on this should be performed in the discussion. How many patients under the age of 10 were included in this cohort? It could be interesting a section on this group

3.2 There should be no interpretations in the results sections. (E.g. comment on the different frequency of genders, 2.2 Majority of SCAs are Late-Onset genetic disorders., 2.3 To best of our knowledge, the coexistence of expansion in SCA1 and SCA2 has not been reported. We have communicated a detailed case report of the two patients with expansion in SCA1 and SCA2 elsewhere). This should be reserved for discussion

3.3 (In section 2.2) It would be also interesting to have the age of onset of the disease, instead of just only the age at examination

3.4 (In section 2.3) the biallelic patients with two CAG repeats of different sizes, but both pathogenic, should be named compound heterozygous, instead of just heterozygous

3.5 Figure 3 legend is not very clear. Is this figure on PMNA, on pathogenic repeats or both?

3.6 (In section 2.5) this section should be exclusively on clinical findings. The diagnostic rate of the different centers is not part of the clinical findings. In fact, I don't see how that is relevant to this paper. Actually, there is no clinical information on this section. It would be important to know the age of onset, type of first symptoms, non-cerebellar symptoms, disease progression...

3.7 (In section 2.6) more clinical information should be provided

Discussion

3.8 The authors discuss the age of onset of the different SCA but they don't present us that data previously... this should be rearranged

3.9 The discussion of patients with biallelic expansions should include not only the references to previously described patients, but also a more detailed discussion of the phenotype of the patients from this cohort

3.10 (In section 3.2) the authors discuss "With the clinical diagnostics data collected for patient's samples, it is observed that phenotypic characteristics may overlap among different SCA types. ", but there is no clinical data presented previously...

3.11 (In section 3.2) "These observations led us to conclude the mimicry of symptoms is more common among SCA1, SCA2 and SCA3 and between SCA2 with SCA12." There is no data supporting this conclusion. The initial diagnosis made by the neurologist should not be taken as a criteria for such a statement. It would be more prudent if the authors exclude this sentence

3.12 Material and methods

The diagnostic approach should be more clearly detailed. From what I could understand of the flowchart, not all patients were submitted to the same diagnostic approach. If so, this is a major limitation of this study and it should be stated as so, as well as explored in the results and discussion

Minor

- Always explain the first time acronyms are use (e.g. TRE)
- In the abstract please order the different SCA by frequency, instead of number
- References: I identified the Ruano reference in duplicate, The authors should review all references

1 st Editorial Decision	31-Jan-2022
Editorial decision: Please resubmit a revised manuscript after addressing all the reviewers' comments	
Editor's understanding of the reviews	
Reviewer #1 Recommends Accept without Revision	
Reviewer #2 Recommends Minor Revision	
Reviewer #3 Recommends Major Revision	

Author's Response to 1 st Review	11-Feb-2022
---	-------------

These are the main reviewer recommendations that the editors believe will make the biggest improvement to this article. **Please do address all reviewer comments listed in the decision letter in your point-by-point response** (you may continue this table to do so if you wish). We hope this summary helps you to understand our decision and expedites the revision process. We value feedback from author and referees alike.
AdvGenet@wiley.com

Reviewer comments	Editor recommendation	Author reply	Changes to Manuscript
2.4 please provide a brief summary of statistical methods as they are not explicitly presented.	ED1 Details of the statistical procedures supporting your Results are essential. Please also briefly mention the statistical tests and significance levels that support results in Tables and Figures.	Thanks, changes have been made as per the suggestions.	Details of Statistical analysis are being added to the experimental methodology under section statistical analysis on Page no. 16
3.7 more clinical information should be provided 2.3 was there any change in genetic testing as time went by and technology and knowledge in the field evolved? Was the genetic panel changed at all over the years? 2.5 Also, please provide the criteria to define "short positive alleles". 3.1 How many patients under the age of 10 were included in this cohort? It could be interesting a section on this group 3.3 It would be also interesting to have the age of onset of the disease 3.6 The diagnostic rate of the different centers.... It would be important to know the age of onset, type of first symptoms, non-cerebellar symptoms, disease progression... 3.9 The discussion of patients with biallelic expansions should include not only the references to previously described patients, but also a more detailed discussion of the phenotype of the patients from this cohort	ED2 Please provide a more thorough and structured account of the clinical procedures, diagnostic criteria and subgroup criteria.	2.3 Yes, Initially the SCA panel consisted of SCA1, SCA2, SCA3, SCA12 (common in India), later with time to expand the diagnostic capabilities, SCA6, SCA7, SCA17 were added to the panel. 2.5 Short positive alleles are normal allele of positive patients. To clarify, "short positive alleles" have been renamed as normal allele of positive patients. 3.1 n=197 cases were under age of 10, Of them 11 cases were diagnosed positive for different SCA loci. 3.7, 3.3 , 3.6, 3.9 Our approach to study SCAs in India is multipronged, and to be able to thoroughly investigate all aspects of this complex disorder to achieve a holistic view we have decided to carry out the investigation in two phases. While phase 1 focuses mainly on genetic screening to enhance the current knowledge repository with an unprecedented sample of around 5000 patients (across different ethnic groups and geographical locations in India), the second phase of the study will focus more on a detailed exploration of clinical findings in an attempt to achieve reliable phenotype-genotype mapping to be able to translate our knowledge of the disorder to clinical settings in the future and achieve robust, reproducible, and consistent results. Currently the patients are referred to us by clinicians upon primary examination. The clinical examination hinges upon collection of patient history and genetic investigation along with tests such as nerve conduction, EMG and AFT forms the primary line of screening. With plans in place to finalize the complementary second phase soon, we attempt to overlay the genetic data we have collected and presented here, with the	2.3 Details have been added to experimental methodology and screening strategy Page no. 16 2.5 Changes have been done short positive alleles" have been renamed as normal allele of positive patients. 3.1 Details added in the results (Page-4) and discussion section (Page-10).

		clinical characteristic and produce a biaxial model to enhance our understanding of the disorder with potential translational benefits	
3.12 The diagnostic approach should be more clearly detailed. From what I could understand of the flowchart, not all patients were submitted to the same diagnostic approach. If so, this is a major limitation of this study and it should be stated as so, as well as explored in the results and discussion	ED3 Please explain the exclusion criteria or difference in diagnostic approach and how this affects the results presented.	3.12 Genetic screening varied for samples with reference to the clinical context. Also, the genetic panel underwent modifications over the years to expand the range of genetic screening offered in accordance with findings relating to SCA prevalence and samples were screened according to the genetic panel available at the time.	Changes have been made in the methodology (page-16) and results section (page-4)
3.10 but there is no clinical data presented previously... 3.11 It would be more prudent if the authors exclude this sentence	ED4 Please check that you have provided the information in the Results that you discuss in the Discussion, cite prior findings or delete these discussion sentences.	3.10 We were referring to the table provided in supplementary data (Table-S1) as clinical diagnostic referrals.	Changes have been made to the manuscript page-14.
2.7 The large difference in frequencies of the most common forms of SCAs is also inevitable, considering the large territorial size of India and the presence of a possible distinct ethnic background for each of these different areas. This topic should be better explored as well, correlating with the ancestral origin in different areas, if this makes sense in the context.	ED5 There are a number of good studies on the demographic history and allele frequencies that would provide background and possibly add clinical utility to this study.	2.7 Yes, the large differences in the frequencies of common SCAs are due to different ethnic populations residing in different areas of India. Like SCA12 is most frequent in a particular community that too in Northern India. But In the manuscript we do not emphasize on the population genetics or ancestral origin , rather we have described our findings for genetic screening of SCAs over a span of 10 years	
2.8 Finally, FRDA is not considered a form of SCA, therefore the inclusion of data on this ARCA should warrant a change in the manuscript's title.	ED6 It is up to the authors whether to change the title, but this excellent point should be discussed and mentioned in the abstract.	2.8 Noted with thanks.	Changes made in title of manuscript and abstract as well.

Reviewer #1

[No specific comments for revision]

Reviewer #2

Sharma et al. present the results of a large, multi-center study of SCAs across India. The authors identified eight different subtypes in the population studied, SCAs 12, 2, 1 and 3 being the most common. The results are important in the context of India and also on the frequency of SCAs among different populations worldwide. Findings on biallelic mutations and regional differences are also remarkable. A few issues need to be addressed prior to publication though.

2.1 English language needs to be reviewed.

Author's reply: Thanks, we have done the changes. The manuscript has been thoroughly reviewed and corrected.

2.2 Please include the meaning of the acronym TRE after it is first mentioned.

Author's reply: Thanks, we have done the changes.

The Introduction is informative and addresses the main data needed to understand the relevance of the study.

2.3 Methodology is adequate, except for a couple of points that require clarification: as the samples were collected and analyzed over a period of 11 years, was there any change in genetic testing as time went by and technology and knowledge in the field evolved? Was the genetic panel changed at all over the years?

Author's reply: Yes, Initially the SCA panel consisted of SCA1, SCA2, SCA3, SCA12 (common in India) for 5 years (2010-2015), later with time to expand the diagnostic capabilities, SCA6, SCA7, SCA17 were added to the panel (2016-2021). We have made changes to the manuscript as well on page no. 16 under screening strategy.

2.4 Second: please provide a brief summary of statistical methods as they are not explicitly presented.

Author's reply: Thanks, details has been added to the statistical analysis in the experimental section (Page 16).

2.5 Also, please provide the criteria to define "short positive alleles".

Author's reply: Sure, short positive alleles are normal alleles of positive patients. To clarify, "short positive alleles" have been renamed as normal allele of positive patients.

2.6 The finding of several cases with biallelic mutations in SCAs is remarkable and findings should be better explored, specially including genotype/phenotype correlation, if available.

Author's reply: Currently the patients are referred to us by clinicians upon primary examination. With plans in place to finalize the complementary second phase soon, we attempt to overlay the genetic data we have collected and presented here.

2.7 The large difference in frequencies of the most common forms of SCAs is also inevitable, considering the large territorial size of India and the presence of a possible distinct ethnic background for each of these different areas. This topic should be better explored as well, correlating with the ancestral origin in different areas, if this makes sense in the context.

Author's reply: Yes, the large differences in the frequencies of common SCAs are due to different ethnic populations residing in different areas of India. Like SCA12 is most frequent in a particular community that too in Northern India. But In the manuscript we do not emphasize on the population genetics or ancestral origin , rather we have described our findings for genetic screening of SCAs over a span of 10 years.

2.8 Finally, FRDA is not considered a form of SCA, therefore the inclusion of data on this ARCA should warrant a change in the manuscript's title.

Author's reply: Noted with thanks, changes have been made to the title of the manuscript and abstract.

Reviewer #3

I would like to start y congratulating the authors on such an interesting work. This is a collaborative work, with an impressive number of patients, and very important conclusion in terms of frequency and characterization of SCAs in India. I agree with the authors with this paper being very important for Indian neurologists, in guiding their practice and genetic testing.

I would just have a few comments/doubts

3.0 The paper would benefit from an overall grammar review. There are some very long sentences, difficult to read. A clear and objective language should be preferred.

Author's reply: Noted with thanks.

Results

3.1 (In section 2.1) I couldn't help noticing a patient with 4 months included in this study. This is very unusual, and a comment on this should be performed in the discussion. How many patients under the age of 10 were included in this cohort? It could be interesting a section on this group

Author's reply: n=197 cases were under age of 10. As we have also included referrals for Autosomal Recessive ataxia diagnosis (FRDA) in our panel, younger aged patients were enrolled. Also Case aged 4 months was a family member of one of the patients referred. Details have been added and also discussed. (Page-10) (Page-4)

3.2 There should be no interpretations in the results sections. (E.g. comment on the different frequency of genders, 2.2 Majority of SCAs are Late-Onset genetic disorders., 2.3 To best of our knowledge, the coexistence of expansion in SCA1 and SCA2 has not been reported. We have communicated a detailed case report of the two patients with expansion in SCA1 and SCA2 elsewhere). This should be reserved for discussion

Author's reply: thanks, done as suggested. Changes have been made to the manuscript on Page-10.

3.3 (In section 2.2) It would be also interesting to have the age of onset of the disease, instead of just only the age at examination

Author's reply: Our approach to study SCAs in India is multipronged, and to be able to thoroughly investigate all aspects of this complex disorder to achieve a holistic view. We have decided to carry out the investigation in two phases. While phase 1 focuses mainly on genetic screening to enhance the current knowledge repository with an unprecedented sample of around 5000 patients (across different ethnic groups and geographical locations in India), the second phase of the study will focus more on a detailed exploration of clinical findings in an attempt to achieve reliable phenotype-genotype mapping to be able to translate our knowledge of the disorder to clinical settings in the future and achieve robust, reproducible, and consistent results. Currently the patients are referred to us by clinicians upon primary examination. The clinical examination hinges upon collection of patient history and genetic investigation along with tests such as nerve conduction, EMG and AFT forms the primary line of screening. With plans in place to finalize the complementary second phase soon, we attempt to overlay the genetic data we have collected and presented here, with the clinical characteristics and produce a biaxial model to enhance our understanding of the disorder with potential translational benefits.

3.4 (In section 2.3) the biallelic patients with two CAG repeats of different sizes, but both pathogenic, should be named compound heterozygous, instead of just heterozygous

Author's reply: Noted, done as suggested on Page -7 of manuscript.

3.5 Figure 3 legend is not very clear. Is this figure on PMNA, on pathogenic repeats or both?

Author's reply: both. Legend has been changed as per suggestion

3.6 (In section 2.5) this section should be exclusively on clinical findings. The diagnostic rate of the different centers is not part of the clinical findings. In fact, I don't see how that is relevant to this paper. Actually, there is no clinical information on this section. It would be important to know the age of onset, type of first symptoms, non-cerebellar symptoms, disease progression...

Author's reply: Addressed in 3.3 comment

3.7 (In section 2.6) more clinical information should be provided

Author's reply: Addressed in 3.3 comment

Discussion

3.8 The authors discuss the age of onset of the different SCA but they don't present us that data previously... this should be rearranged

Author's reply: thanks, changes have been made as suggested. (Page -10)

3.9 The discussion of patients with biallelic expansions should include not only the references to previously described patients, but also a more detailed discussion of the phenotype of the patients from this cohort

Author's reply: Addressed in 3.3 comment

3.10 (In section 3.2) the authors discuss "With the clinical diagnostics data collected for patient's samples, it is observed that phenotypic characteristics may overlap among different SCA types. ", but there is no clinical data presented previously...

Author's reply: We were referring to the table provided in supplementary data (Table-S1) as clinical diagnostic referrals. Changes have been made to the manuscript (page-14).

3.11 (In section 3.2) "These observations led us to conclude the mimicry of symptoms is more common among SCA1, SCA2 and SCA3 and between SCA2 with SCA12." There is no data supporting this conclusion. The initial diagnosis made by the neurologist should not be taken as a criteria for such a statement. It would be more prudent if the authors exclude this sentence.

Author's reply: After neurological consultations patients were diagnosed with another SCA type which was discordant with genetic testing results. The basis for reported clinical mimicry was the clinician's opinion supported by findings from ancillary testing. As previously mentioned, the primary examination involves tests including but not limited to EMG and AFT. The clinician's experience substantiated by the findings from these tests led us to suspect the existence of a particular SCA in a patient but results from genetic testing were incongruous. This led us to believe that there is more significant overlap in clinical phenotypes among certain SCAs than others. The sentence has been more clearly re-framed.(Page-14)

3.12 Material and methods

The diagnostic approach should be more clearly detailed. From what I could understand of the flowchart, not all patients were submitted to the same diagnostic approach. If so, this is a major limitation of this study and it should be stated as so, as well as explored in the results and discussion

Author's reply: Genetic screening varied for samples with reference to the clinical context. Also, the genetic panel underwent modifications over the years to expand the range of genetic screening offered in accordance with findings relating to SCA prevalence and samples were screened according to the genetic panel available at the time. Changes have been made in results (Page-4) and screening strategy (Page-16).

Minor

- Always explain the first time acronyms are use (e.g. TRE)

Author's reply: this has been addressed

- In the abstract please order the different SCA by frequency, instead of number

Author's reply: this has been addressed

- References: I identified the Ruano reference in duplicate, The authors should review all references

Author's reply: this has been addressed

Final Decision	11-Feb-2022
----------------	-------------

The authors have addressed the reviewers' comments and the article is accepted for publication.